# On Leave-One-Out Conditional Mutual Information For Generalization

**Mohamad Rida Rammal**
University of California, Los Angeles
ridarammal@g.ucla.edu

**Alessandro Achille**
Caltech, AWS AI Labs
aachille@caltech.edu

**Aditya Golatkar**
University of California, Los Angeles
aditya29@cs.ucla.edu

**Suhas Diggavi**
University of California, Los Angeles
suhas@ee.ucla.edu

**Stefano Soatto**
University of California, Los Angeles
soatto@ucla.edu

## Abstract

We derive information theoretic generalization bounds for supervised learning algorithms based on a new measure of leave-one-out conditional mutual information (loo-CMI). In contrast to other CMI bounds, which may be hard to evaluate in practice, our loo-CMI bounds are easier to compute and can be interpreted in connection to other notions such as classical leave-one-out cross-validation, stability of the optimization algorithm, and the geometry of the loss-landscape. It applies both to the output of training algorithms as well as their predictions. We empirically validate the quality of the bound by evaluating its predicted generalization gap in scenarios for deep learning. In particular, our bounds are non-vacuous on image-classification tasks.

## 1 Introduction

Generalization in classical machine learning models is often understood in terms of the bias-variance trade-off: over-parameterized models fit the data better but are more likely to overfit. However, the push for ever larger deep network models, driven by their remarkable empirical generalization, has spurred a race for the development of new theories of generalization with two main objectives: (1) guiding the practitioners in designing over-parameterized architectures and training schemes that would generalize better and (2) providing provable bounds on the performance of the model after deployment.

In this context, information theoretic bounds on generalization [27, 31, 5, 21, 29, 12, 16] are particularly appealing. They capture the intuitive idea that models that memorize less about the training set will generalize better. In particular, since they deal with the amount of *information* stored in the weights rather than the *number* of weights, they are especially adaptable to over-parameterized models [17, 2, 1]. However, the bounds are often vacuous in realistic settings, require modified training algorithms, or are difficult to compute and/or to relate to relevant properties of the network/training algorithm. To remedy this problem, we ask whether it is possible to develop a bound that is: (a) realistically tight for standard training algorithms used in deep learning; (b) interpretable, meaning that they can be rewritten in terms of quantities the practitioners can control during the

36th Conference on Neural Information Processing Systems (NeurIPS 2022).

training process and that may guide toward the design of better training algorithms, (c) that applies out-of-the-box to standard training algorithms.

**Contributions.** As a step in this direction, we introduce a new information theoretic bound based on *leave-one-out conditional mutual information* (loo-CMI)[1]. The main intuition behind the bound is to ask the counterfactual question "If we were to remove one sample at random from the training set and retrain, how well would we be able to infer which sample was removed?". As we shall see, bounding this quantity is enough to bound sharply the amount of memorization in a deep network, and hence its generalization (Theorems 3.1 and 3.2). At the same time, the bound is easy to interpret and connects to several empirical quantities and other generalization theories, such as: (1) stability of the optimization algorithm (Section 4.4), (2) flatness of the loss landscape at convergence (Section 4.5). Moreover, we show that our bound can also be interpreted in the function space (that is, looking at the activations rather than the weights, (Theorem 3.2) leading to tighter bounds for over-parameterized models through the data processing inequality.

Our loo-CMI bound falls in the general class of conditional mutual information bounds, introduced by [29]. However, while standard CMI bounds iterate over all the subset of samples of size $N$ ($2^N$ subsets for a dataset of $2N$ samples) in order to compute the bound, our loo-CMI bounds only need to iterate over $N$ possible samples to remove, leading to an exponential reduction in the cost to estimate the bound. Moreover, our strategy of removing a single sample is easy to relate to the stability of the training path to slight perturbation of the training set, thus giving a clear connection between generalization and the geometry of the loss landscape (Figure 1). Empirically, we show that our bound can be computed and is non-vacuous for state-of-the-art deep networks fine-tuned on standard image classification tasks (Table 1). We also study the dependency of the bound on the size of the dataset (Figure 2), and the hyper-parameters (Figure 3).

## 2 Preliminaries

We denote with $[n]$ the set $\{1, \ldots, n\}$ and with $\mathbb{R}^d$ the set of real-valued and complex-valued $d-$dimensional vectors. $A^T$ and $A^{-1}$ denote the transpose and inverse of a square matrix $A$. If $x$ is a vector with $n$ elements, then $\mathrm{diag}(x)$ denotes the $n \times n$ square matrix with the elements of $x$ on its main diagonal; $\|\cdot\|$ is the standard Euclidean norm on vectors. We use uppercase letters to denote random variables, and lowercase letters to denote a specific value the random variable can take. If $X$ is a random variable, then $p_X$ is the probability distribution of $X$, and $\mathbb{E}_{x \sim p_X}[x]$ denotes the expected value of $X$. For a pair of random variables $X$ and $Y$, $p_{(X,Y)}$ and $p_X p_Y$ are the joint and product distributions, respectively. If $p$ and $q$ are two probability densities over $\mathbb{R}^d$ such that $p$ is absolutely continuous with respect to $q$ ($p \ll q$), the Kullback-Leibler Divergence from $p$ to $q$ is defined as:

$$\mathrm{KL}(p \parallel q) = \int_{\mathbb{R}^d} p(x) \log\left(\frac{p(x)}{q(x)}\right). \tag{1}$$

The Mutual Information between two random variables $X$ and $Y$, and the conditional mutual information of $X$ and $Y$ given a third random variable $Z$ are respectively given by:

$$I(X;Y) = \mathrm{KL}(p_{(X,Y)} \parallel p_X p_Y). \tag{2}$$

$$I(X;Y \mid Z) = \mathbb{E}_{z \sim P_Z}[\underbrace{I(X;Y \mid z)}_{I(X;Y|Z=z)}]. \tag{3}$$

### 2.1 Problem Formulation

Let $Z^n = \{Z_1, \ldots, Z_n\} \in \mathcal{Z}^n$ be a collection of $n$ independent and identically distributed samples drawn from some unknown distribution $\mathcal{P}$. We consider the setting of supervised learning where $\mathcal{Z} = \mathcal{X} \times \mathcal{Y}$; i.e., each sample is composed of a feature vector and a label. Let $\mathcal{A} : \mathcal{Z}^n \to \mathcal{W} \subseteq \mathrm{R}^K$ be a possibly stochastic training algorithm, where $\mathcal{W}$ is a set of weights (e.g., weights in a neural network). Given a loss function $\ell : \mathcal{W} \times \mathcal{Z} \to \mathbb{R}_+$, the empirical risk for weights $w$ on $Z^n$ is given

---

[1]loo-CMI was also independently introduced in [11]. This work was not available at the time of our submission; it was pointed to us along with the paper decision. To our understanding, loo-CMI was introduced in [11] for mainly studying the generalization properties of interpolating algorithms.

by $\widehat{\mathcal{L}}(w, Z^n) = 1/n \sum_{i=1}^{n} \ell(w, Z_i)$, and the "true" loss is given by $\mathcal{L}(w) = \mathbb{E}_{Z' \sim \mathcal{P}} [\ell(w, Z')]$; i.e., the average loss $w$ incurs on random samples drawn from $\mathcal{P}$.

Our goal is to find a $w^* \in \mathcal{W}$ such that $w^* = \arg\min_{w \in \mathcal{W}} \mathcal{L}(w)$. Since we do not have access to the data-generating distribution $\mathcal{P}$, computing the true loss is infeasible. However, we do have access to a set of samples $Z^n$, and one approach is empirical risk minimization where we find the $w \in \mathcal{W}$ that minimizes $\widehat{\mathcal{L}}(w, Z^n)$. If the difference $\mathcal{L}(w) - \widehat{\mathcal{L}}(w, Z^n)$ is small, then the true loss of an empirical risk minimizer would be nearly optimal within the considered hypothesis class. Therefore, it is of great interest to study the generalization gap of $\mathcal{A}$ (given $n - 1$ samples for ease of notation):

$$\text{gen}(\mathcal{A}) = \left| \mathbb{E}_{\mathcal{A}, z^{n-1} \sim \mathcal{P}^{n-1}} \left[ \mathcal{L}(\mathcal{A}(z^{n-1})) - \widehat{\mathcal{L}}(\mathcal{A}(z^{n-1}), z^{n-1}) \right] \right|. \tag{4}$$

## 2.2 Conditional Mutual Information Bounds

We start by recalling the main results on Conditional Mutual Information (CMI) bounds that we return to later. Let $\tilde{Z}^{2n} \in \mathcal{Z}^{n \times 2}$ be a dataset with $2n$ samples drawn from a distribution $\mathcal{P}$, grouped into $n$ pairs. Let $S \sim \mathcal{S} = \text{Uniform}(\{0,1\}^n)$ a uniform random binary vector of size $n$. $S$ selects one sample from each pair in $\tilde{Z}^{2n}$ to form $\tilde{Z}_S^n \in \mathcal{Z}^n$ given by $(\tilde{Z}_S^n)_i = \tilde{Z}_{S_i+1}^{2n}$. In this setting, the bound of Steinke and Zakynthinou [29] arises from asking the following question : "if a model is trained with a subset of samples chosen through the random binary indexing vector $S$, how much information does the output of the algorithm provide about $S$?" Here, the Shannon Mutual Information is useful because it is a measure of the amount of 'information' obtained about one random variable after observing another random variable.

**Theorem 2.1** (Theorem 2, [29]). *Let $\ell : \mathcal{W} \times \mathcal{Z} \to [0,1]$ be a bounded loss function. Let $\mathcal{A}$ be a possibly stochastic training algorithm. Let $W = \mathcal{A}(\tilde{Z}_S^n)$ be the output of $\mathcal{A}$ given the dataset $\tilde{Z}_S^n$, then*

$$\text{gen}(\mathcal{A}) \leq \sqrt{\frac{2}{n} I(W; S \mid \tilde{Z}^{2n})}. \tag{5}$$

Haghifam et al. [12] improved Theorem 2.1 by moving the expectation over $\tilde{Z}^{2n}$ outside the square root, and by measuring the information the output of the algorithm provides on random *subsets* of the indexing vector $S$. Harutyunyan et al. [16] further improved the bound by moving the expectation over the random subsets outside the square root.

**Theorem 2.2** (Theorem 2.6, [16]). *Let $\ell : \mathcal{W} \times \mathcal{Z} \to [0,1]$ be a bounded loss function. Let $m \in [n]$ and let $V \sim \mathcal{V}$ be a random subset of $[n]$ of size $m$. Let $W = \mathcal{A}(\tilde{Z}_S)$ be the output of $\mathcal{A}$ given the dataset $\tilde{Z}_S$, then*

$$\text{gen}(\mathcal{A}) \leq \mathbb{E}_{\tilde{z}^{2n} \sim \mathcal{P}^{2n}, v \sim \mathcal{V}} \sqrt{\frac{2}{m} I(W; S_v \mid \tilde{z}^{2n})}. \tag{6}$$

One appealing aspect of the CMI bounds is that they work for any training algorithm and any model, including over-parameterized networks trained with Stochastic Gradient Descent (SGD). However, computing the bounds requires iterating over all possible $2^N$ values of $S$. Moreover, the bound is difficult to interpret since the value of $W = \mathcal{A}(\tilde{Z}_S^n)$ for different values of $S$ can vary significantly and in unpredictable ways for large non-convex models. Taking inspiration from leave-one-out cross validation, we propose a different CMI bound which avoids both problems by removing a single sample from the dataset. As we shall see, the bound allows for interpretable expressions, faster computation, and a connection to notions of stability.

## 3 Leave-One-Out Conditional Mutual Information

Let $Z^n = \{Z_1, Z_2, \ldots, Z_n\} \in \mathcal{Z}^n$ be a dataset of $n$ i.i.d. samples drawn from $\mathcal{P}$. Let $U \sim \mathcal{U} = \text{Uniform}([n])$ be uniform random variable taking values over the indices of the samples in $Z$. $U$ removes a single sample from $Z^n$ to form $Z_{-U}^{n-1} = Z^n \setminus Z_U$, the dataset without the $U^{\text{th}}$ sample. Let $W = \mathcal{A}(Z_{-U}^{n-1})$ be the output of the algorithm, then we define the pointwise leave-one-out conditional mutual information as:

$$\text{loo-CMI}(\mathcal{A}, z^n) = I(W; U \mid z^n). \tag{7}$$

In this setting, loo-CMI$(\mathcal{A}, z^n)$ measures the amount of information that the output of the algorithm $W$ reveals about $U$, the index (identity) of the left-out sample. As is the case with the conditional mutual information terms in Theorems 2.1 and 2.2, loo-CMI is bounded. Specifically, loo-CMI is upper bounded by the entropy of $U$, $H(U) = \log(n)$. Hence, it does not suffer from the same issues as information stability bounds [27, 31]. Moreover, the output of an algorithm is not significantly affected by the inclusion or exclusion of one sample when the input consists of thousands of other samples, so we expect loo-CMI to be small for large values of $n$. We use (7) to derive a bound on the generalization of error in expectation (Theorem 3.1).

## 3.1 Generalization Bounds Based on loo-CMI

loo-CMI is deeply intertwined with leave-one-out cross validation, so before we derive bounds on the generalization through loo-CMI, it is helpful to first obtain a probabilistic upper bound on the leave-one-out cross validation error (loo-cv). loo-cv measures the difference in loss between the samples which the algorithm trained on, and the sample that was left out. We begin with a precise definition of loo-cv and a probabilistic bound on it.

$$\text{loo-cv}(w, z^n, u) = \frac{1}{n-1} \sum_{i \neq u} \ell(w, z_i) - \ell(w, z_u). \tag{8}$$

**Lemma 3.1.** *Let* $\ell : \mathcal{W} \times \mathcal{Z} \to [0, 1]$ *be a bounded loss function, then for all* $t > 0$, $w \in \mathcal{W}$, *and* $z^n \in \mathcal{Z}$,

$$\mathbb{E}_{u \sim \mathcal{U}} \left[ \exp \left( t \cdot (\text{loo-cv}(w, z^n, u)) \right) \right] \leq \exp \left( \frac{t^2 c_n^2}{8} \right), \qquad c_n = \frac{n}{n-1}. \tag{9}$$

To derive the generalization bounds included in this paper, we make heavy use of Lemma 3.1. We begin with a generalization bound in expectation.

**Theorem 3.1.** *Let* $\mathcal{A} : \mathcal{Z}^{n-1} \to \mathcal{W}$ *be a training algorithm. Let* $\ell : \mathcal{W} \times \mathcal{Z} \to [0, 1]$ *be a bounded loss function, then*

$$\text{gen}(\mathcal{A}) \leq \frac{c_n}{\sqrt{2}} \mathbb{E}_{z^n \sim \mathcal{P}^n} \sqrt{\text{loo-CMI}(\mathcal{A}, z^n)}, \tag{10}$$

where $c_n$ is given in (9). The bound in (10) guarantees a good generalization error when loo-CMI$(\mathcal{A}, z^n)$ is small, i.e., when the output of the algorithm $W$ tells us little about the identity of the sample that was removed from the training dataset. If a parametric learning algorithm ends up memorizing the samples in the training set, the generalization gap could be large, and so would loo-CMI as we can determine $U$ by checking which sample was not memorized by the algorithm. On the other extreme, a learning algorithm which outputs a constant parameter regardless of the training set does not have a generalization gap, and in this case loo-CMI would equal 0 as $W$ provides no information about $U$. We note that the bound of Theorem 3.1 most closely resembles the bound of Theorem 2.2 when $m = 1$. The latter bound is given by:

$$\text{gen}(\mathcal{A}) \leq \frac{1}{n} \sum_{i=1}^{n} \mathbb{E}_{\tilde{z}^{2n} \sim \mathcal{P}^{2n}} \sqrt{2I(W; S_i \mid \tilde{z}^{2n})}. \tag{11}$$

Recalling the setting of Section 2.2, (11) is computed by first going through every possible value of $S \in \{0, 1\}^n$. For every value of $S$ and for each $i \in [n]$, one fixes $S = (S_1, S_2, \ldots, S_n)$ excluding $S_i$ which varies uniformly over $\{0, 1\}$. In other words, the $i^{\text{th}}$ component of the training set $\tilde{Z}_S^n$ is equally likely to be either $\tilde{Z}_{i,1}$ or $\tilde{Z}_{i,2}$. The right-hand-side of (11) then bounds the generalization using the information that the weights contain about $S_i$. In other words, the question asked is: "can the output of the algorithm help us determine which one of $\tilde{z}_{i,1}$ or $\tilde{z}_{i,2}$ was present in the training set?". On the other hand, we ask a different question: "can the output of the algorithm help us determine the index of the sample that was removed?". This subtle change leads to an exponential reduction in the cost of computing the since we only need to iterate over the values of $U \in [n]$, as opposed to $S$ which can take $2^n$ possible values.

## 3.2 Extension To The Function Space

The previous bounds used the output of the algorithm, the weights, to bound the generalization error. A different approach studied in [16] is to assume that given a training dataset and a test

sample, the algorithm outputs a prediction on the test sample. In other words, we assume the algorithm is a possibly stochastic function $h : \mathcal{Z}^{n-1} \times \mathcal{X} \longrightarrow \mathcal{R}$ which takes a training set $z$, a new unlabeled sample $x_{\text{new}}$, and outputs a possibly stochastic prediction $h(z, x_{\text{new}})$ on the new sample. The set of predictions $\mathcal{R}$ can be different than the set of labels $\mathcal{Y}$ (e.g., class probabilities in multi-class classification tasks). Unlike bounds with respect to the weights, this approach is applicable to both parametric algorithms (e.g. neural networks) and non-parametric algorithms (e.g. $k$-nearest neighbors). In the former case, the output of the algorithm $\mathcal{A}$ specifies a prediction function $g \in \{g_w : \mathcal{X} \to \mathcal{R} \mid w \in \mathcal{W}\}$ from a class of functions parameterized by the weights i.e. $h(z, x_{\text{new}}) = g_{\mathcal{A}(z)}(x_{\text{new}})$.

We redefine the loss to be a non-negative function $\ell : \mathcal{R} \times \mathcal{Y} \to \mathbb{R}_+$ that measures the distance between the predictions and the ground-truth labels. Given $\ell$ and a training set $Z^n$, the true loss of an algorithm $h$ is given by $\mathcal{L}(h, Z^n) = \mathbb{E}_{(x', y') \sim \mathcal{P}} [\ell(h(Z^n, x'), y')]$, and the empirical loss is given by $\widehat{\mathcal{L}}(h, Z^n) = 1/n \sum_{i=1}^{n} \ell(h(Z^n, X_i), Y_i)$. The generalization gap of $h$ given is then defined as:

$$\text{gen}(h) = \left| \mathbb{E}_{h, z^{n-1} \sim \mathcal{P}^{n-1}} \left[ \widehat{\mathcal{L}}(h, z^{n-1}) - \mathcal{L}(h, z^{n-1}) \right] \right|. \tag{12}$$

Recalling the setup of Section 3, we define the pointwise functional leave-one-out conditional mutual information floo-CMI as:

$$\text{floo-CMI}(h, z^n) = I \left( h \left( z_{-U}^{n-1}, x^n \right); U | z^n \right). \tag{13}$$

While loo-CMI measures the reduction in uncertainty about $U$ after having known the weights, floo-CMI measures the reduction in uncertainty after having known the predictions made on the *whole* dataset (including the prediction on the removed sample). We derive a bound with respect to floo-CMI$(h, z^n)$ using a proof that closely resembles the proof of Theorem 3.1.

**Theorem 3.2.** *Let $\ell : \mathcal{R} \times \mathcal{Y} \to [0, 1]$ be a bounded loss function. Let $U$ be a uniform random variable over $[n]$, then*

$$\text{gen}(h) \leq \frac{c_n}{\sqrt{2}} \mathbb{E}_{z^n \sim \mathcal{P}^n} \sqrt{\text{floo-CMI}(h, z^n)}, \tag{14}$$

where $c_n$ is given in (9). For a parametric algorithm $\mathcal{A}$ with prediction function $h$, $U$—$\mathcal{A}(z_{-U}^{n-1})$—$h(z_{-U}^{n-1}, X)$ is a Markov chain. By the data-processing inequality, we have that floo-CMI$(h, z^n) \leq$ loo-CMI$(\mathcal{A}, z^n)$. Since the bounds of Theorems 3.1 and 3.2 differ only through loo-CMI and floo-CMI, the bound of Theorem 3.2 is sharper.

# 4 Computing The Bounds

## 4.1 Computable Upper Bounds to loo-CMI and floo-CMI

To evaluate and interpret the bounds for a parametric algorithm $\mathcal{A}$ with prediction function $h$, we require a computable closed-form expressions for loo-CMI and floo-CMI. However, obtaining a closed form expression is a difficult task in most cases. To see the reason behind this difficulty, note that loo-CMI can be written as:

$$\text{loo-CMI}(\mathcal{A}, z^n) = I(W; U \mid z^n) = \mathbb{E}_{u \sim \mathcal{U}} \left[ \text{KL}(p_{W \mid z^n, u} \| p_{W \mid z^n}) \right], \tag{15}$$

where $p_{W \mid z^n} = \mathbb{E}_{u \sim \mathcal{U}} \left[ p_{W \mid z^n, u} \right]$ is a mixture distribution. Since one of the terms in the Kullback-Leibler divergence of (15) is a mixture distribution, it is unlikely that one could find a closed-form expression for the mutual information except for simple cases (e.g., we cannot obtain a closed-expression for Gaussian mixtures[8]). To find a computable and interpretable bound on the generalization error, we propose upper bounds on the conditional mutual information which can be interpreted and evaluated.

**Theorem 4.1.** *Let $\mathcal{A}$ be a parametric algorithm with prediction function $h$. Let $U, Z^n, W$ be defined as before. Let $U'$ be an identical independent version of $U$ and $W' = \mathcal{A}(z_{U'}^{n-1})$, then*

$$\text{loo-CMI}(\mathcal{A}, z^n) \leq -\mathbb{E}_{u \sim \mathcal{U}} \left[ \ln \mathbb{E}_{u' \sim \mathcal{U}} \left[ \exp \left( -\text{KL}(p_{W \mid z^n, u} \| p_{W' \mid z^n, u'}) \right) \right] \right], \tag{16}$$

$$\text{floo-CMI}(h, z^n) \leq -\mathbb{E}_{u \sim \mathcal{U}} \left[ \ln \mathbb{E}_{u' \sim \mathcal{U}} \left[ \exp \left( -\text{KL}(p_{h(z_{-u}^{n-1}, x)} \| p_{h(z_{-u'}^{n-1}, x)}) \right) \right] \right]. \tag{17}$$

One can alternatively upper bound loo-CMI through the convexity of the Kullback-Leibler divergence (similarly for floo-CMI). However, since the function $(-\ln)$ is strictly convex, it is easy to show that the bound of Theorem 4.1 is strictly tighter through Jensen's inequality. Moreover, we opt to use this bound as it allows for more interpretable expressions (Corollary 4.1).

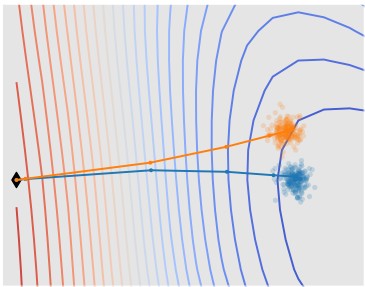

Figure 1: **Visualization of the quantity involved in the bound.** We plot a projection of the loss landscape (contour lines) of a ResNet-18 trained on MIT-67 using the technique of [20]. In orange and blue we show the projection of training paths obtained by removing two different samples from the dataset. Equation (18) bounds the test error as a function of the distance between the final points of the two paths, normalized by amount of noise added (displayed by the cloud around the final weights). Note that if the loss landscape is flat we can add more noise, therefore obtaining a better bound (Section 4.5). If the optimization algorithm is stable, the two paths will remain close at all times, which will also improve the final bound (Section 4.4).

## 4.2 Geometry Aware Synthetic Randomization

Theorems 3.1 and 3.2 are valid in theory, but the outputs of algorithms used in practice are deterministic given a fixed input, and so we do not have a distribution of weights or predictions. This means that (7) and (13) are degenerate. Even if we vary the random seed for neural networks trained with SGD, and run the algorithm for each dataset and random seed, we obtain a set of discrete distributions with unequal supports. We alleviate this issue by taking measures similar to the ones made in [12, 22, 15], and also previously in related contexts in [2, 9, 1, 23]. In particular, we add noise to the output of a deterministic algorithm, and use the now stochastic algorithm to bound the information in the weights and predictions. We avoid adding isotropic Gaussian noise for the weight-based bounds, as changing the values of some weights may have little effect on the loss compared to others. Therefore, we add noise while taking into account the geometry of the loss landscape.

Specifically, let $\mathcal{A}$ be deterministic algorithm, and let $\mathcal{A}_\Sigma(z_{-U}^{n-1}) = \mathcal{A}(z_{-U}^{n-1}) + N$, where $N \sim \mathcal{N}(\mathbf{0}, \Sigma)$. A choice of $\Sigma$ that would incorporate the notion that weights have a varying effect on the loss is $\Sigma = \mathrm{diag}(\alpha_1, \ldots, \alpha_K)$ with $\alpha_k > 0$ for all $k$ and $\alpha_i$ not necessarily equal to $\alpha_j$ for $i \neq j$. Similarly, we can add noise to predictions made by a deterministic algorithm as $h_\sigma(z_{-U}^{n-1}, x) = h(z_{-U}^{n-1}, x) + M$, where $M \sim \mathcal{N}(\mathbf{0}, \sigma^2 I)$. The generalization bounds derived for $\mathcal{A}_\Sigma$ or $h_\sigma$ are not directly applicable to $\mathcal{A}$ and $h$, but if certain Lipschitz continuity assumptions hold for the loss function $\ell$, it is possible to derive bounds for deterministic algorithms from the bounds of their noisy counterparts (Theorems 4.2 and 4.3). We begin by applying Theorem 4.1 to get interpretable expressions for the conditional mutual information terms.

**Corollary 4.1.** *Let $\mathcal{A}$ be a deterministic algorithm with prediction function $h$. Let $w_{-i} = \mathcal{A}(z_{-i}^{n-1})$ and $h_{-i} = h(z_{-i}^{n-1}, x^n)$ be the output and predictions respectively when $i^{\text{th}}$ sample is removed from the training set. Using Theorem 4.1, we obtain*

$$\text{loo-CMI}(\mathcal{A}_\Sigma, z^n) \leq \ln(n) - \frac{1}{n} \sum_{i=1}^{n} \ln \left( \sum_{j=1}^{n} e^{-\frac{1}{2}(w_{-i} - w_{-j})^T \Sigma^{-1} (w_{-i} - w_{-j})} \right), \qquad (18)$$

$$\text{floo-CMI}(h_\sigma, z^n)) \leq \ln(n) - \frac{1}{n} \sum_{i=1}^{n} \ln \left( \sum_{j=1}^{n} e^{-\frac{1}{2}\|h_{-i} - h_{-j}\|^2/\sigma^2} \right). \qquad (19)$$

## 4.3 Connections to Stability and Bounds For Deterministic Algorithms

It is easy to see the connection between classical definitions of stability and the right-hand side of (18), (19). After all, to compute the right-hand side of (18) and (19), one only needs to know how the weights and predictions of the algorithm change when two datasets differ by one sample. Based on

the observation that components of the weight might have differing degrees of effect on the loss, and the right-hand side of (18), we introduce the idea of "relative" weight stability. Moreover, we also use classical definitions of functional stability, and here we find it useful and intuitive to define two notions of stability: one with respect to the samples that the two datasets share, and one with respect to any other sample. The definitions are as follows.

**Definition 4.1.** *Let $z^{n-1}, \hat{z}^{n-1} \in \mathcal{Z}^n$ be datasets such that $z^{n-1}$ and $\hat{z}^{n-1}$ differ by at most one sample. Letting $a = \mathcal{A}(z^{n-1}) - \mathcal{A}(\hat{z}^{n-1})$, then we say that a deterministic algorithm $\mathcal{A}$ has $\epsilon$ weight stability relative to positive semi-definite matrix $\Sigma$ if $a^T \Sigma^{-1} a \leq \epsilon^2$.*

**Definition 4.2.** *Let $z^{n-1}, \hat{z}^{n-1} \in \mathcal{Z}^n$ be datasets such that $z^{n-1}$ and $\hat{z}^{n-1}$ differ by at most one sample, and without loss of generality, assume they differ at the first sample i.e. $z_1 \neq \hat{z}_1$ and $z_k = \hat{z}_k$ for all $k \neq 1$. Let $h$ be prediction function $h : \mathcal{Z}^{n-1} \times \mathcal{X} \to \mathbb{R}^d$, then we say $h$ has $\beta$-train stability if $\left\| h(z^{n-1}, z_k) - h(\hat{z}^{n-1}, z_k) \right\|^2 \leq \beta^2$ for all $k \neq 1$. Moreover, we say that $h$ has $\beta_1$-test stability if $\left\| h(z^{n-1}, x') - h(\hat{z}^{n-1}, x') \right\|^2 \leq \beta_1^2$ for all $x' \in \mathcal{X}$.*

Given bounds on the noisy version of a deterministic algorithm, one can derive bounds on the deterministic algorithm itself by making certain Lipschitz continuity assumptions on the loss function. Moreover, we show that given relative weight stability and functional stability of deterministic algorithm, we can add an optimal amount of noise to find a bound on the deterministic algorithm. The following technique is also used by [16, 22] for the case of isotropic noise. We generalize this result for arbitrary values of the noise covariance.

**Theorem 4.2.** *Let $\mathcal{A} : \mathcal{Z}^{n-1} \to \mathcal{W}$ be a deterministic algorithm, and let $\ell : \mathcal{W} \times \mathcal{Z} \to \mathbb{R}_+$ be the loss that a set of weights incur on a sample. If $\ell(w, \cdot)$ is L-Lipschitz in the weights, then if $\mathcal{A}$ has $\epsilon$-weight stability relative to a positive semi-definite $\Sigma$, then $\operatorname{gen}(\mathcal{A}) \leq \sqrt{4 c_n \epsilon L \sqrt{\operatorname{tr}(\Sigma)}}$.*

**Theorem 4.3.** *Let $h : \mathcal{Z}^n \times \mathcal{X} \to \mathbb{R}^d$ be a deterministic prediction function. Assume that $h$ has $\beta$-train stability and $\beta_1$-test stability. Assume the loss function $\ell : \mathcal{R} \times \mathcal{Y}$ is L-Lipschitz continuity in the first coordinate, then $\operatorname{gen}(\mathcal{A}) \leq \sqrt{4 c_n L \sqrt{nd(n\beta^2 + 2\beta_1^2)}}$.*

### 4.4 Bounds For Stochastic Gradient Descent

As we have seen, the quality of our information bound depends on the stability of the optimization algorithm. Stochastic Gradient Descent is the most commonly used method in deep learning, and it is therefore interesting to study the stability of SGD and how it translates into generalization from an information theoretic point of view. Assuming the gradient updates are $\gamma$-bounded, $\|w_{t+1} - w_t\| \leq \sqrt{\gamma}$ for $\gamma > 0$ and all $t > 0$, where $w_t$ are the weights at iteration $t$, then one can bound $\|w_{-i} - w_{-j}\|$ with respect to the number of iterations of SGD [14], and obtain a bound on the generalization through the right-hand-side of (18). In particular, we derive the following bound which show how training for a shorter time $T$ and having more bounded updates $\gamma$ both contribute to generalization. The bound relates to the observation that early stopping (training for a smaller number of steps T) may improve the generalization of the network.

**Lemma 4.1.** *Let the gradient update rule be $\gamma$-bounded. Suppose we run SGD for $T$ steps, and we use the same starting value for the weights, then $\operatorname{gen}(\mathcal{A}_{\sigma^2 I}) \leq \sqrt{\frac{c_n^2 T^2 \gamma}{\sigma^2}}$.*

### 4.5 Local interpretation of the bound

The bound in Corollary 4.1 depends on non-local quantities as it requires retraining the model a linear number of times on subsets of the data. Ideally, one wants to bound the generalization without having to retrain. We now provide a qualitative approximation of the bound using local quantities, in order to connect the bound with the geometry of the loss function. Assume that the training algorithm minimizes the loss $\mathcal{A}(z^n) = \arg\min_{w \in \mathcal{W}} \ell(z^n, w)$, and let $w^* = \mathcal{A}(z^n)$ be the minimum to which the algorithm converges, and similarly let $w^*_{-i} = \mathcal{A}(z^{n-1}_{-i})$. Using influence functions [19], we can approximate:

$$w^*_{-i} - w^* \approx \frac{1}{n} H_{w^*}^{-1} \nabla_w \ell(z_i, w^*) = g_i, \tag{20}$$

where $H_w$ is the Hessian of the loss, $\nabla_w \ell(z_i, w^*)$ is the gradient of the loss on $z_i$ for $w^*$, and we have introduced the per-sample gradient at convergence rescaled by the Hessian $g_i = \frac{1}{n} H_{w^*}^{-1} \nabla_w \ell(z_i, w^*)$.

Using this notation we can approximate:

$$\text{loo-CMI}(\mathcal{A}_\Sigma, z) \leq \ln(n) - \frac{1}{n} \sum_{i=1}^{n} \ln \left( \sum_{j=1}^{n} e^{-\frac{1}{2}(w_{-i} - w_{-j})^T \Sigma^{-1}(w_{-i} - w_{-j})} \right), \qquad (21)$$

$$\approx \ln(n) - \frac{1}{n} \sum_{i=1}^{n} \ln \left( \sum_{j=1}^{n} e^{-\frac{1}{2}(g_i - g_j)^T \Sigma^{-1}(g_i - g_j)} \right), \qquad (22)$$

$$\overset{(*)}{=} \ln(n) - \frac{1}{n} \sum_{i=1}^{n} \ln \left( \sum_{j=1}^{n} e^{-\frac{1}{2}(g_i - g_j)^T H_{w^*}(g_i - g_j)} \right), \qquad (23)$$

where in $(*)$ we have assumed $\Sigma^{-1} = H_{w^*}$, which is the optimal variance of the noise (Section 4.2).

From (23) we see that converging to a flat minimum, that is, having a small norm of the Hessian $H_{w^*}$, is expected to correlate to better generalization. This is in accordance with several empirical results connecting flat minima to better generalization [18, 9, 6]. It should be noted that flatness by itself cannot explain generalization, since we can reparameterize the network to have identical predictions (and hence generalization) but arbitrarily large Hessian [7]. Indeed, in (23) we see that the role of the Hessian is mediated by the similarity of the (re-scaled) per-sample gradients at convergence, which can change under reparameterization. In particular, studying flatness by itself is not enough.

## 5    Related Work

Over the past several years, there has been a significant line of research in using information theory both to interpret the behavior of DNNs [2, 28, 1] and to derive generalization bounds [27, 31, 5, 21, 29, 12, 16], with the earliest works on this line from [3, 27, 31]. Many of them develop information stability bounds which involve the mutual information between the output of the algorithm and the samples [27, 31, 5, 21] which could degenerate, *e.g.,* if the data is continuous. This was observed in [29], which then proposed a new bound based on the conditional mutual information (see Theorem 2.1) which is non-degenerate. This work was further extended by [12, 16]. In particular [16] developed bounds based on prediction outputs rather than the algorithm outputs, and its combination with the conditional mutual information framework made it the state-of-the-art in terms of information-theoretic generalization bounds. The two issues identified in our work related to this is one of computability (as the [16] bound might in principle require computation exponential in the size of the dataset) and interpretability; this is the main focus of our work. In particular, our loo-CMI framework can not only be computed more efficiently, but also connects to classical leave-one-out cross validation measures used extensively in practice (see Theorem 3.1). Another aspect introduced in [16, 22] is the application of CMI bounds to deterministic algorithm outputs (as one can only run a finite number of runs of even a randomized algorithm on a dataset). We use this viewpoint in our work, but using a geometry-aware method (see Theorem 4.2), which takes into account the loss landscape in the algorithmic output space. There has also been a line of work connecting SGD to generalization including earlier works in [13] and more recently its connection with information theoretic bounds [25, 22]. We apply these ideas to the framework of loo-CMI (see Lemma 4.1). The connection to classical notions of algorithmic stability can also be made using the loo-CMI framework (see Theorem 4.3).

## 6    Experiments

**Model and datasets.**    We now study the behavior of our loo-CMI and floo-CMI bounds on real-world image classification tasks. In particular, we fine-tune an off-the-shelf ResNet-18 model pretrained on ImageNet on a set of standard image-classification tasks (see also Table 1): MIT-67 [26] and Oxford Pets [24] with a few thousand examples each. On both datasets we fine-tune for 10 epochs using stochastic gradient descent with learning rate $\eta = 0.05$, momentum $m = 0.99$, batch size $B = 256$, and weight decay $\lambda = 0.0005$. We compute $w_{-i}/h_{-i}$ by removing sample $i$ from the training set and re-training from scratch. For all the experiments we remove 10 samples one at a time across 3 random seeds and use corollary 4.1 to compute the information bounds. We used 2 NVIDIA 1080Ti GPUS and the experiments take approximately 1-2 days.

| Dataset | Train Error | Test Error | loo-CMI | floo-CMI |
|---------|-------------|------------|---------|----------|
| MIT-67 [26] | 0 | 0.339 | 9.65 | 0.493 |
| Ox. Pets [24] | 0 | 0.134 | 8.51 | 0.514 |

Table 1: **Generalization bounds on image-classification tasks.** We report the train and test error (fraction of wrong predictions) and our generalization bounds on several image classification dataset ($\sigma = 0.1$).

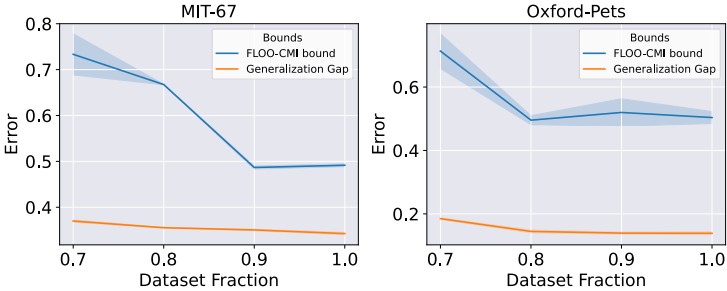

Figure 2: floo-CMI **bound and dataset size**. For different datasets, we plot of the floo-CMI bound as the fraction of samples used in training increases. We observe that larger datasets have better generalization bounds. This is expected as stability improves with larger datasets.

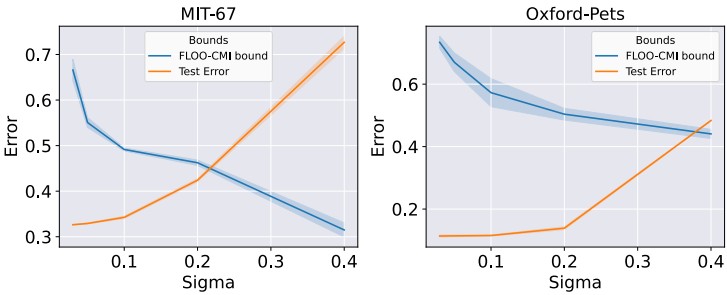

Figure 3: **Effect of varying $\sigma$.** Increasing the value of $\sigma$ in (14) improves the floo-CMI bound (blue line) but also makes $h_\sigma$ increasingly different from $h$ (as shown by the increased test error, orange line), thus making bounding the generalization of the original algorithm $h$ given $h_\sigma$ more difficult.

**Non-vacuous bounds.** In Table 1, we compute the loo-CMI and floo-CMI bounds using (10) and (14). We show that while the former bound is vacuous, the floo-CMI bound provides non-vacuous generalization bounds on all datasets. This is significant, as obtaining non-vacuous bound for large-scale models remains a challenging problem. The failure of the bound based on loo-CMI is expected here, since loo-CMI looks directly at the information contained in the weights of the network without considering how this information is used. In large models with millions of parameters, most of the information in the weights do not significantly affect the predictions and should ideally be discarded. This is indeed done by the floo-CMI bound.

**Effect of the dataset size.** To show how the quality of the bound changes for different sizes of the dataset, we subsample randomly and without replacement $n$ samples from each dataset. In Figure 2, we plot the resulting generalization gap alongside our generalization bounds as the size of the subsample varies (in terms of fractions of the dataset). We observe that the generalization bound becomes tighter as the size of the training set grows. This is not surprising as we expect the model to become more stable for larger datasets.

**Synthetic randomization.** In Section 4.2, we introduced artificial Gaussian noise $N(0, \Sigma)$ in order to obtain better bounds. Increasing the noise variance $\Sigma$ in the loo-CMI bound, or $\sigma$ in floo-CMI, improves the bound on the CMI, but also increases the test error of the model. Hence, we need to

select a value of $\Sigma$ and $\sigma$ that provides a good trade-off between the two. In Figure 3, we show how changing the noise variance affects each term.

## 7  Conclusion

In this work, we introduced a bound based on a new information-theoretic measure, *i.e.,* leave-one-out conditional mutual information and analyzed its properties. loo-CMI bounds reduce the computational costs compared to prior CMI bounds, connects to several empirical quantities and generalization theories. It leads to non-vacuous bounds for several deep learning applications in image classification. There are several open questions including, whether it is possible to (i) obtain tighter bounds using the leave-one-out framework for general learning algorithms; (ii) further reduce the computational cost of computing loo-CMI[2] and (iii) obtain deeper understanding the effect of loss landscape and dynamics of learning algorithms on learning performance using the information-theoretic lens.

## Acknowledgments and Disclosure of Funding

This work was supported in part by NSF grants #1955632, #2146838, #2139304, #2007714, ONR N00014-22-1-2252, and ARO W911NF-17-1-0304. The views and conclusions contained in this document are those of the authors and should not be interpreted as representing the official policies, either expressed or implied, of the Army Research Office or the U.S. Government. The U.S. Government is authorized to reproduce and distribute reprints for Government purposes notwithstanding any copyright notation herein.

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
