# A  Proofs

## A.1  Proof of Lemma 1

We begin by restating the result of Lemma 3.1.

**Lemma.** *Let $\ell : \mathcal{W} \times \mathcal{Z} \to [0,1]$ be a bounded loss function, then for all $t > 0$, $w \in \mathcal{W}$, and $z^n \in \mathcal{Z}$,*

$$\mathbb{E}_{u \sim \mathcal{U}} \left[ \exp \left( t \cdot (\text{loo-cv}(w, z^n, u)) \right) \right] \leq \exp \left( \frac{t^2 c_n^2}{8} \right), \qquad c_n = \frac{n}{n-1}. \tag{24}$$

**Proof**: To show this, we use tools from [30]. Let $\ell \in [0,1]^n$ and $t > 0$. For $u \in [n]$, we define:

$$f(\ell, u) = \frac{1}{n-1} \sum_{i \neq u} \ell_i - \ell_u. \tag{25}$$

If we obtain a bound on $\mathbb{E}_{u \sim \mathcal{U}} \left[ \exp(t \cdot f(\ell, u)) \right]$ for every $\ell \in [0,1]^n$, then we also obtain a bound on $\mathbb{E}_{u \sim \mathcal{U}} \left[ \exp \left( t \cdot (\text{loo-cv}(w, z^n, u)) \right) \right]$ for all $z^n \in \mathcal{Z}^n, w \in \mathcal{W}$. Now,

$$\mathbb{E}_{u \sim \mathcal{U}} \left[ \exp(t \cdot f(\ell, u)) \right] = \frac{1}{n} \exp \left( \frac{t}{n-1} \sum_{i=1}^n \ell_i \right) \sum_{j=1}^n \exp \left( -c \ell_j \right) =: \frac{1}{n} g(\ell). \tag{26}$$

where $c := t c_n = t(1 + \frac{1}{n-1})$. Since the maximum of $g$ must be at a stationary point, then

$$\frac{\partial g(\ell)}{\partial \ell_j} = \frac{t}{n-1} \exp \left( \frac{t}{n-1} \sum_{i=1}^n \ell_i \right) \left( \sum_{i=1}^n e^{-c \ell_i} - n e^{-c \ell_j} \right), \tag{27}$$

Setting (27) to 0, then the right-hand side expression of (27) must be 0. This implies that $\ell_j$ is either at the extremities or we must have a value of $\ln(b)/c$, where

$$b = \frac{1}{n} \sum_{i=1}^n e^{-c \ell_i}. \tag{28}$$

Let $m$ be the number of $\ell_j$ that are 0, and $k$ the number of $\ell_j$ that are 1. Solving for $b$ using (28), we get that

$$b = \frac{m + k e^{-c}}{m + k}. \tag{29}$$

We find that $b$ depends on $\ell$ only through the number of 0s and 1s in $\ell$. Writing $g(\ell)$ with respect to $m, k$ and $b$, we obtain

$$g(\ell) = n \exp \left( \frac{t}{n-1} k + \frac{m+k}{n} \ln(b) \right), \tag{30}$$

Hence, finding the maximum of $g(\ell)$ is equivalent to finding the solution for the following optimization problem.

$$\max_{m, k \geq 0} \quad \frac{t}{n-1} k + \frac{m+k}{n} \ln(b) \tag{31}$$
$$\text{s.t.} \quad m + k \leq n.$$

Since $m$ and $k$ are bounded non-negative integers, we can find the solution of the above maximization through a brute-force search. However, we can relax the constraints of (31), and allow $m, k$ to take non-integer values over $[0, n]$. Using the method of Lagrange multipliers, we have that:

$$\min_{\lambda \geq 0} \max_{m, k \geq 0} L(m, k, \lambda) = \min_{\lambda \geq 0} \max_{m, k \geq 0} \left( \frac{t}{n-1} k + \frac{m+k}{n} \ln(b) - \lambda(m + k - n) \right). \tag{32}$$

The partial derivative of (32) with respect to $m$ and $k$ are given by:

$$\nabla_m L = \frac{k(1 - e^{-c})}{n(m + k e^{-c})} + \frac{\ln(b)}{n} - \lambda, \tag{33}$$

$$\nabla_k L = \frac{t}{n-1} - \frac{m(1 - e^{-c})}{n(m + k e^{-c})} + \frac{\ln(b)}{n} - \lambda. \tag{34}$$

Setting the partial derivatives of (33) and (34) to 0 yields

$$m = \frac{1 - e^{-c} - ce^{-c}}{c - 1 + e^{-c}} k \tag{35}$$

$$m + k = \left( \frac{c(1 - e^{-c})}{c - 1 + e^{-c}} - 1 \right) k. \tag{36}$$

Substituting these expressions in (32), we see that we now need to find

$$\max_{k \geq 0} \quad \frac{t}{n - 1} k \left( 1 + \frac{(1 - e^{-c})}{c - 1 + e^{-c}} \ln \left( \frac{1 - e^{-c}}{c} \right) \right)$$

$$\text{s.t.} \quad k \leq \frac{c - 1 + e^{-c}}{c(1 - e^{-c})} n.$$

Since $1 + \frac{(1 - e^{-c})}{c - 1 + e^{-c}} \ln \left( \frac{1 - e^{-c}}{c} \right) > 0$ for all $c > 0$, this expression is maximized by the largest value $k$ can take. Hence, the we obtain our maximum for

$$k^* = \left( -1 + \frac{c}{c(1 - e^{-c})} \right) n,$$

$$m^* = n - k,$$

$$b^* = \frac{1 - e^{-c}}{c}.$$

Plugging into our expression, we have

$$g(\ell) \leq n \exp \left( -1 + \frac{c}{1 - e^{-c}} + \log \left( \frac{1 - e^{-c}}{c} \right) \right), \tag{37}$$

Recalling that $\mathbb{E}_{u \sim \mathcal{U}} \left[ \exp(t \cdot f(\ell, u)) \right] = 1/ng(\ell)$, we obtain

$$\mathbb{E}_{u \sim \mathcal{U}} \left[ \exp(t \cdot f(\ell, u)) \right] \leq \exp \left( -1 + \frac{c}{1 - e^{-c}} + \log \left( \frac{1 - e^{-c}}{c} \right) \right) \tag{38}$$

Using the Taylor expansion of the exponent in the right-hand side of (38), we have that:

$$-1 + \frac{c}{1 - e^{-c}} + \log \left( \frac{1 - e^{-c}}{c} \right) \leq \frac{c^2}{8}. \tag{39}$$

Finally, we obtain

$$\mathbb{E}_{u \sim \mathcal{U}} \left[ \exp \left( t \cdot (\text{loo-cv}(w, z^n, u)) \right) \right] \leq \mathbb{E}_{u \sim \mathcal{U}} \left[ \exp(t \cdot f(\ell, u)) \right] \tag{40}$$

$$\leq \exp(\frac{c^2}{8}). \tag{41}$$

Recalling that $c = tc_n$, we obtain the desired result.

## A.2 Proof of Theorem 3.1

We begin restating the Theorem.

**Theorem.** *Let $\mathcal{A} : \mathcal{Z}^{n-1} \to \mathcal{W}$ be a training algorithm. Let $\ell : \mathcal{W} \times \mathcal{Z} \to [0, 1]$ be a bounded loss function, then*

$$\text{gen}(\mathcal{A}) \leq \frac{c_n}{\sqrt{2}} \mathbb{E}_{z^n \sim \mathcal{P}^n} \sqrt{\text{loo-CMI}(\mathcal{A}, z^n)}, \tag{42}$$

We state a series of lemmas and definitions that are essential for the proof.

**Lemma A.1.** *Let $z^n \in \mathcal{Z}^n$, and let $w \in \mathcal{W}$, then*

$$\mathbb{E}_{u \sim \mathcal{U}} \left[ \text{loo-cv}(z^n, w, u) \right] = 0. \tag{43}$$

*Proof.* We have that:

$$n \cdot \mathbb{E}_{u \sim \mathcal{U}} \left[ \text{loo-cv}(z^n, w, u) \right] = \sum_{u=1}^{n} \left( \frac{1}{n-1} \sum_{i \neq u} \ell(w, z_i) - \ell(w, z_u) \right), \tag{44}$$

$$= \frac{1}{n-1} \sum_{u=1}^{n} \sum_{i \neq u} \ell(w, z_i) - \sum_{u=1}^{n} \ell(w, z_u), \tag{45}$$

$$= \frac{1}{n-1} \sum_{i=1}^{n} (n-1)\ell(w, z_i) - \sum_{u=1}^{n} \ell(w, z_u), \tag{46}$$

$$= 0. \tag{47}$$

$\square$

**Lemma A.2.** *Let $\mathcal{A} : \mathcal{Z}^{n-1} \to \mathcal{W}$ be a training algorithm, then*

$$\text{gen}(\mathcal{A}) = \left| \mathbb{E}_{\mathcal{A}, z^n \sim \mathcal{P}^n, u \sim \mathcal{U}} \left[ \text{loo-cv}(z^n, \mathcal{A}(z_{-u}^{n-1}), u) \right] \right| \tag{48}$$

*Proof.*

$$\text{gen}(\mathcal{A}) = \left| \mathbb{E}_{\mathcal{A}, z^{n-1} \sim \mathcal{P}^{n-1}} \left[ \mathcal{L}(\mathcal{A}(z^{n-1})) - \widehat{\mathcal{L}}(\mathcal{A}(z^{n-1}), z^{n-1}) \right] \right|, \tag{49}$$

$$= \left| \mathbb{E}_{\mathcal{A}, z^{n-1} \sim \mathcal{P}^{n-1}} \left[ \mathbb{E}_{z' \sim \mathcal{P}} \left[ \mathcal{L}(\mathcal{A}(z^{n-1}), z') \right] - \widehat{\mathcal{L}}(\mathcal{A}(z^{n-1}), z^{n-1}) \right] \right|, \tag{50}$$

$$= \left| \mathbb{E}_{\mathcal{A}, z^n \sim \mathcal{P}^n, u \sim \mathcal{U}} \left[ \mathcal{L}(\mathcal{A}(z_{-u}^{n-1}), z_u) - \widehat{\mathcal{L}}(\mathcal{A}(z_{-u}^{n-1}), z_{-u}^{n-1}) \right] \right|, \tag{51}$$

$$= \left| \mathbb{E}_{\mathcal{A}, z^n \sim \mathcal{P}^n, u \sim \mathcal{U}} \left[ \text{loo-cv}(z^n, u, \mathcal{A}(z_{-u}^{n-1}))) \right] \right|. \tag{52}$$

$\square$

**Definition A.1.** *We say that a random variable $X \sim P_X$ is $\sigma^2-$subgaussian if:*

$$\mathbb{E}_{x \sim P_X} \left[ t\exp(x - \mathbb{E}_{x \sim P_x}[x]) \right] \leq \exp \left( \frac{t^2 \sigma^2}{2} \right). \tag{53}$$

**Lemma A.3** (Lemma 1, [31]). *Let $A, B$ be arbitrary random variables. Let $A', B'$ be independent copies of $A, B$ such that $P_{A', B'} = P_{A'} P_{B'}$. Suppose $f(A', B')$ is $\sigma^2$-subgaussian, then*

$$\left| \mathbb{E}_{A, B} \left[ g(A, B) - \mathbb{E}_{A', B'} \left[ g(A', B') \right] \right] \right| \leq \sqrt{2\sigma^2 I(A; B)}. \tag{54}$$

Now, fix $Z^n = z^n$, and let $A = \mathcal{A}(z_{-U}^{n-1})$ and $B = U$, and let

$$f(A, B) = \text{loo-cv}(z^n, A, B). \tag{55}$$

Using Lemma A.1, $E_{A', B'} f(A', B') = 0$. Moreover, we use $f(A', B')$ is $\sigma^2$-subgaussian with $\sigma^2 = c_n^2/4$ using Lemma 3.1. This gives us that:

$$\left| \mathbb{E}_{\mathcal{A}, u \sim \mathcal{U}} \left[ \text{loo-cv}(z^n, \mathcal{A}(z_{-u}^{n-1}), u) \right] \right| \leq \sqrt{\frac{c_n^2}{2} I(W; U \mid z^n)}, \tag{56}$$

where $W = \mathcal{A}(z_{-U}^{n-1})$. Taking the expectation over $z^n$ on both sides, we get that:

$$\mathbb{E}_{z^n \sim \mathcal{P}^n} \left| \mathbb{E}_{\mathcal{A}, u \sim \mathcal{U}} \left[ \text{loo-cv}(\mathcal{A}(z_{-u}^{n-1}), z^n, u) \right] \right| \leq \mathbb{E}_{z^n \sim \mathcal{P}^n} \sqrt{\frac{c_n^2}{2} I(W; U \mid z^n)}. \tag{57}$$

Since $g(\cdot) = |\cdot|$ is a convex function, then

$$\text{gen}(\mathcal{A}) = \left| \mathbb{E}_{\mathcal{A}, z^n \sim \mathcal{P}^n u \sim \mathcal{U}} \left[ \text{loo-cv}(\mathcal{A}(z_{-u}^{n-1}), z^n, u) \right] \right| \tag{58}$$

$$\leq \mathbb{E}_{z^n \sim \mathcal{P}^n} \left| \mathbb{E}_{\mathcal{A}, u \sim \mathcal{U}} \left[ \text{loo-cv}(\mathcal{A}(z_{-u}^{n-1}), z^n, u) \right] \right| \tag{59}$$

$$\leq \frac{c_n}{\sqrt{2}} \mathbb{E}_{z^n \sim \mathcal{P}^n} \sqrt{I(W; U \mid z^n)}. \tag{60}$$

## A.3 Proof of Theorem 3.2

We begin by restating the Theorem.

**Theorem.** *Let $\ell : \mathcal{R} \times \mathcal{Y} \to [0,1]$ be a bounded loss function. Let $U$ be a uniform random variable over $[n]$, then*

$$\text{gen}(h) \leq \frac{c_n}{\sqrt{2}} \mathbb{E}_{z^n \sim \mathcal{P}^n} \sqrt{\text{floo-CMI}(h, z^n)}, \tag{61}$$

This proof is nearly identical to the proof of Theorem 3.1. Here, we use the prediction function instead of the weights. We use the alternate definition of the loss $\ell : \mathcal{R} \times \mathcal{Y} \to [0,1]$. Let $g \in \mathcal{R}^n$ be a set of prediction, then we also use an alternate definition of the leave-one-out cross validation where

$$\text{loo-cv}(z^n, g, u) = \frac{1}{n-1} \sum_{i \neq u} \ell(g_i, y_i) - \ell(g_u, y_u). \tag{62}$$

**Lemma A.4.** *Let $z^n \in \mathcal{Z}^n$, and let $g \in \mathcal{R}^n$, then*

$$\mathbb{E}_{u \sim \mathcal{U}} \left[ \text{loo-cv}(z^n, g, u) \right] = 0. \tag{63}$$

**Lemma A.5.** *Let $h : \mathcal{Z}^{n-1} \times \mathcal{X} \to \mathcal{R}$ be a training algorithm, then*

$$\text{gen}(\mathcal{A}) = \left| \mathbb{E}_{h, z^n \sim \mathcal{P}^n, u \sim \mathcal{U}} \left[ \text{loo-cv}(z^n, h(z_{-u}^{n-1}, x^n), u) \right] \right|. \tag{64}$$

Now, we use Lemma A.3 with $A = h(z_{-u}^{n-1}, x^n)$, $B = U$, and

$$f(A, B) = \text{loo-cv}(z^n, A, B). \tag{65}$$

The proof then follows in the exact same way as it did for Theorem 3.1, and we get the desired result:

$$\text{gen}(h) \leq \frac{c_n}{\sqrt{2}} \mathbb{E}_{z^n \sim \mathcal{P}^n} \sqrt{I(h(z_{-U}^{n-1}, x^n); U \mid z^n)}. \tag{66}$$

## A.4 Proof of Theorem 4.1

We begin by restating the Theorem.

**Theorem.** *Let $\mathcal{A}$ be a parametric algorithm with prediction function $h$. Let $U, Z^n, W$ be defined as before. Let $U'$ be an identical independent copy of $U$, then*

$$\text{loo-CMI}(\mathcal{A}, z^n) \leq -\mathbb{E}_{u \sim \mathcal{U}} \left[ \ln \mathbb{E}_{u' \sim \mathcal{U}} \left[ \exp \left( -\text{KL}(p_{W \mid z^n, u} \parallel p_{W \mid z^n, u'}) \right) \right] \right], \tag{67}$$

$$\text{floo-CMI}(h, z^n) \leq -\mathbb{E}_{u \sim \mathcal{U}} \left[ \ln \mathbb{E}_{u' \sim \mathcal{U}} \left[ \exp \left( -\text{KL}(p_{h(z_{-u}^{n-1}, x)} \parallel p_{h(z_{-u'}^{n-1}, x)}) \right) \right] \right]. \tag{68}$$

We begin by proving a more general result. Let $W \sim P_W, U \sim P_U$ be arbitrary random variables. Suppose $w \in \mathcal{W}$ and $u \in \mathcal{U}$. Let $q(u')$ be any probability distribution over $\mathcal{U}$, then we define

$$\Phi(q(u'), u) = \int_{\mathcal{W}} p(w|u) \int_{\mathcal{U}} q(u') \ln \left( \frac{p(w|u)q(u')}{p(w|u')p(u')} \right) \mathrm{d}u' \mathrm{d}w. \tag{69}$$

We begin with the following lemma.

**Lemma A.6.**

$$I(W; U) \leq \mathbb{E}_{u \sim P_U} \left[ \Phi(q, u) \right]. \tag{70}$$

*Proof.* We note that

$$\Phi(q, u) = \int_{\mathcal{W}} p(w|u) \int_{\mathcal{U}} q(u') \ln \left( \frac{p(w|u)q(u')}{p(w|u')p(u')} \right) \mathrm{d}u' \mathrm{d}w \tag{71}$$

$$= -\int_{\mathcal{W}} p(w|u) \int_{\mathcal{U}} q(u') \ln \left( \frac{p(w|u')p(u')}{p(w|u)q(u')} \right) \mathrm{d}u' \mathrm{d}w. \tag{72}$$

$$= -\int_{\mathcal{W}} p(w|u) \mathbb{E}_{u' \sim q_U} \left[ \ln \left( \frac{p(w|u')p(u')}{p(w|u)q(u')} \right) \right] \mathrm{d}w. \tag{73}$$

$$\geq -\int_{\mathcal{W}} p(w|u) \ln \left( \mathbb{E}_{u' \sim q_U} \left[ \frac{p(w|u')p(u')}{p(w|u)q(u')} \right] \right) \mathrm{d}w, \tag{74}$$

where the last inequality follows from the concavity of $\ln$ and Jensen's inequality. Continuing from (74), we obtain

$$\Phi(q, u) \geq - \int_{\mathcal{W}} p(w|u) \ln \left( \mathbb{E}_{u' \sim P_U} \left[ \frac{p(w|u')p(u')}{p(w|u)q(u')} \right] \right) \mathrm{d}w \tag{75}$$

$$= - \int_{\mathcal{W}} p(w|u) \ln \left( \int_{\mathcal{U}} q(u') \frac{p(w|u')p(u')}{p(w|u)q(u')} \mathrm{d}u' \right) \mathrm{d}w \tag{76}$$

$$= - \int_{\mathcal{W}} p(w|u) \ln \left( \int_{\mathcal{U}} \frac{p(w|u')p(u')}{p(w|u)} \mathrm{d}u' \right) \mathrm{d}w \tag{77}$$

$$= - \int_{\mathcal{W}} p(w|u) \ln \left( \frac{p(w)}{p(w|u)} \right) \mathrm{d}w \tag{78}$$

$$= \mathrm{KL}(p(w|u) \parallel p(w)). \tag{79}$$

Since $I(W; U) = \mathbb{E}_{u \sim P_U} [\mathrm{KL}(p(w|u) \| \| p(w))]$, we obtain the result of the lemma. $\qquad \square$

To find a tight bound, we minimize $\Phi(q(u'), u)$ with respect to the probability distribution $q(u')$. Since $\phi(q(u', u)$ is an upper bound to $\mathrm{KL}(p(w|u) \| \| p(w))$ for any $q(u')$, the infimum of $\Phi$ with respect to $q(u')$ is still an upper bound. Since we are trying to find the infimum with respect to a probability distribution, the Lagrangian is given by

$$\tilde{\Phi}(q(u')) = \int_{\mathcal{W}} p(w|u) \int_{\mathcal{U}} q(u') \ln \left( \frac{p(w|u)q(u')}{p(w|u')p(u')} \right) \mathrm{d}u' \mathrm{d}w + \lambda \left( \int_{\mathcal{U}} q(u') \mathrm{d}u' - 1 \right). \tag{80}$$

We "differentiate" with respect to $q(u')$, and obtain

$$\frac{\partial \tilde{\Phi}(q(u'))}{\partial q(u')} = \int_{\mathcal{W}} p(w|u) \ln \left( \frac{p(w|u)q(u')}{p(w|u')p(u')} \right) \mathrm{d}w + 1 + \lambda. \tag{81}$$

Setting (81) to 0, and solving for $q(u')$ for every $u' \in \mathcal{U}$, we obtain

$$q^*(u') = \frac{p(u')e^{-\mathrm{KL}(p(w|u)\|\|p(w|u'))}}{\int_{\mathcal{U}} p(\hat{u})e^{-\mathrm{KL}(p(w|u)\|\|p(w|\hat{u}))} \mathrm{d}\hat{u}} \tag{82}$$

To find the expression for $\Phi(q^*(u'), u)$, note that:

$$\mathrm{KL}(q^*(u') \parallel p(u')) = - \ln \left( \int_{\mathcal{U}} p(u')e^{-\mathrm{KL}(p(w|u) \parallel p(w|u'))} \mathrm{d}u' \right) - \int_{\mathcal{U}} q^*(u')\mathrm{KL}(p(w|u) \parallel p(w|u')) \mathrm{d}u'. \tag{83}$$

Plugging in the above into $\Phi(q^*(u'), u)$, we obtain

$$\Phi(q^*(u'), u) = \int_{\mathcal{U}} q^*(u')\mathrm{KL}(p(w|u) \parallel p(w|u')) \mathrm{d}u' + \mathrm{KL}(q^*(u') \parallel p(u')) \tag{84}$$

$$= - \ln \left( \int_{\mathcal{U}} p(u')e^{-\mathrm{KL}(p(w|u) \parallel p(w|u'))} \mathrm{d}u' \right) \tag{85}$$

$$= - \ln \mathbb{E}_{u' \sim P_U} \left[ e^{-\mathrm{KL}(p(w|u) \parallel p(w|u'))} \right] \tag{86}$$

Finally, taking the expectation with respect to $U$ and using Lemma A.6, we obtain that

$$I(W; U) \leq -\mathbb{E}_{u \sim P_U} \left[ - \ln \mathbb{E}_{u' \sim P_U} \left[ \mathrm{KL}(p_{W \mid u} \parallel p_{W \mid u'}) \right] \right]. \tag{87}$$

We apply the above to $I(W; U|z^n)$ and $I(h(z_{-i}^{n-1}, x^n), U \mid z^n)$ to obtain the expressions of Theorem 4.1.

## A.5  Proof of Theorem 4.2

We begin a restatement of the theorem and the following lemma:

**Theorem.** *Let $\mathcal{A} : \mathcal{Z}^{n-1} \to \mathcal{W}$ be a deterministic algorithm, and let $\ell : \mathcal{W} \times \mathcal{Z} \to \mathbb{R}_+$ be the loss that a set of weights incur on a sample. If $\ell(w, \cdot)$ is L-Lipschitz in the weights, then if $\mathcal{A}$ has $\epsilon$-weight stability relative to a positive semi-definite $\Sigma$, then $\mathrm{gen}(\mathcal{A}) \leq \sqrt{4c_n \epsilon L \sqrt{\mathrm{tr}(\Sigma)}}.$*

**Lemma A.7.** *Let $\mathcal{A} : \mathcal{Z}^{n-1} \to \mathcal{W}$ be a deterministic algorithm Let the loss $\ell : \mathcal{W} \times \mathcal{Z} \to \mathbb{R}_+$ be L-Lipschitz in the weights. Let $\Sigma$ be a positive definite matrix, then*

$$\mathrm{gen}(\mathcal{A}) \le \mathrm{gen}(\mathcal{A}_\Sigma) + 2L\sqrt{\mathrm{tr}(\Sigma)}. \tag{88}$$

*Proof.* We have that:

$$\mathrm{gen}(\mathcal{A}) = \left| \mathbb{E}_{\mathcal{A}, z^n \sim \mathcal{P}^n} \left[ \mathcal{L}(\mathcal{A}(z^n)) - \widehat{\mathcal{L}}(\mathcal{A}(z^n), z^n) \right] \right|. \tag{89}$$

$$= \left| \mathbb{E}_{\mathcal{A}, z^n \sim \mathcal{P}^n, z' \in \mathcal{P}} \left[ \ell(\mathcal{A}(z^n), z') - \frac{1}{n} \sum_{i=1}^n \ell(\mathcal{A}(z^n), z_i) \right] \right|. \tag{90}$$

$$= \left| \mathbb{E}_{\mathcal{A}, z^n \sim \mathcal{P}^n, z' \in \mathcal{P}} \left[ \ell(\mathcal{A}_\Sigma(z^n), z') + \delta' - \frac{1}{n} \sum_{i=1}^n \left( \ell(\mathcal{A}_\Sigma(z^n), z_i) + \delta_i \right) \right] \right|, \tag{91}$$

$$\le \left| \mathbb{E}_{\mathcal{A}, z^n \sim \mathcal{P}^n, z' \in \mathcal{P}} \left[ \ell(\mathcal{A}_\Sigma(z^n), z') - \frac{1}{n} \sum_{i=1}^n \left( \ell(\mathcal{A}(z^n), z_i) \right) \right] \right| + |\delta'| + \frac{1}{n} \sum_{i=1}^n |\delta_i|, \tag{92}$$

$$\le \mathrm{gen}(\mathcal{A}_\Sigma) + \mathbb{E}_{\mathcal{A}, z^n \sim \mathcal{P}^n, z' \in \mathcal{P}} \left[ |\delta'| + \frac{1}{n} \sum_{i=1}^n |\delta_i| \right], \tag{93}$$

where $\delta' = \ell(\mathcal{A}(z^n), z') - \ell(\mathcal{A}_\Sigma(z^n), z')$ and $\delta_i = \ell(\mathcal{A}(z^n), z_i) - \ell(\mathcal{A}_\Sigma(z^n), z_i)$. Recalling that $\mathcal{A}_\Sigma(z^n) = \mathcal{A}(z^n) + N$ where $N \sim \mathcal{N}(0, \Sigma)$, we have that $|\delta'| \le L \|N\|$ (also for $|\delta_i|$). This gives us that:

$$\mathrm{gen}(\mathcal{A}) \le \mathrm{gen}(\mathcal{A}_\Sigma) + \mathbb{E}_{\mathcal{A}, z^n \sim \mathcal{P}^n, z' \in \mathcal{P}} \left[ |\delta'| + \frac{1}{n} \sum_{i=1}^n |\delta_i| \right], \tag{94}$$

$$= \mathrm{gen}(\mathcal{A}_\Sigma) + 2L \|N\|, \tag{95}$$

$$\le \mathrm{gen}(\mathcal{A}_\Sigma) + 2L\sqrt{\mathrm{tr}(\Sigma)}, \tag{96}$$

where the last inequality follows from the fact that for $N \sim \mathcal{N}(0, \Sigma)$, $\mathbb{E}[\|N\|] \le \mathrm{tr}(\Sigma)$. $\square$

Now, suppose $\mathcal{A}$ has $\epsilon$-weight stability relative to $\Sigma$, and let $\alpha > 0$, then using (10) and (18), we have

$$\mathrm{gen}(\mathcal{A}_{\alpha^2 \Sigma}) \le \frac{c_n \epsilon}{2\alpha}. \tag{97}$$

Hence, for all $\alpha > 0$, we have that:

$$\mathrm{gen}(\mathcal{A}) \le \frac{c_n \epsilon}{2\alpha} + 2L\alpha\sqrt{\mathrm{tr}(\Sigma)}. \tag{98}$$

Finding the minimum of the above with respect to $\alpha$, we obtain:

$$\mathrm{gen}(\mathcal{A}) \le 2\sqrt{\frac{c_n \epsilon}{2} \cdot 2L\sqrt{\mathrm{tr}(\Sigma)}}, \tag{99}$$

$$= \sqrt{4 c_n \epsilon L \sqrt{\mathrm{tr}(\Sigma)}}. \tag{100}$$

## A.6 Proof of Theorem 4.3

We begin with a restatement of the theorem, and the following lemma where proof is identical to the lemma used in Section A.5.

**Theorem.** *Let $h : \mathcal{Z}^n \times \mathcal{X} \to \mathbb{R}^d$ be a deterministic prediction function. Assume that $h$ has $\beta$-train stability and $\beta_1$-test stability. Assume the loss function $\ell : \mathcal{R} \times \mathcal{Y}$ is L-Lipschitz continuity in the first coordinate, then $\mathrm{gen}(\mathcal{A}) \le \sqrt{4 c_n L \sqrt{nd(n\beta^2 + 2\beta_1^2)}}$.*

**Lemma A.8.** *Let $h : \mathcal{Z}^{n-1} \times \mathcal{X} \to \mathcal{R} \subset \mathbb{R}^d$ be a deterministic prediction function. Let the loss $\ell : \mathcal{R} \times \mathcal{Y} \to \mathbb{R}_+$ be L-Lipschitz in the predictions. Let $\sigma^2$ be a positive definite matrix, then*

$$\mathrm{gen}(h) \le \mathrm{gen}(h_\sigma) + 2L\sigma\sqrt{nd}. \tag{101}$$

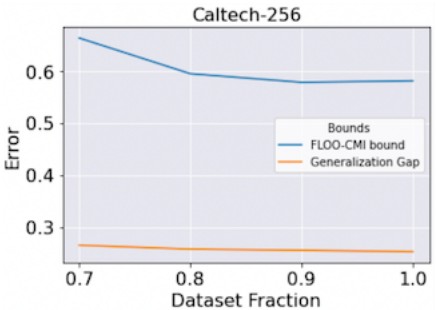

Figure 4: floo-CMI **bound and dataset size**. For Caltech256.

Now, for a prediction function $h$ with $\beta$-train stability and $\beta_1$-test stability, we have that

$$\text{gen}(h_\sigma) \le \frac{c_n\sqrt{(n-2)\beta^2 + 2\beta_1^2}}{2\sigma} \le \frac{c_n\sqrt{n\beta^2 + 2\beta_1^2}}{2\sigma}. \tag{102}$$

Hence, we have that

$$\text{gen}(h) \le \frac{c_n\sqrt{n\beta^2 + 2\beta_1^2}}{2\sigma} + 2L\sigma\sqrt{nd}. \tag{103}$$

Optimizing the right-hand-side of the above over $\sigma$, we obtain the desired result:

$$\text{gen}(h) \le \sqrt{4c_n L \sqrt{nd(n\beta_1 + 2\beta_1^2)}}. \tag{104}$$

### A.7 Proof of Lemma 4.1

We restate the lemma.

**Lemma.** *Let the gradient update rule be $\gamma$-bounded. Suppose we run SGD for $T$ steps, and we use the same starting value for the weights, then* $\text{gen}(\mathcal{A}_{\sigma^2 I}) \le \sqrt{\frac{c_n^2 T^2 \gamma}{\sigma^2}}$.

**Definition A.2** (Definition 2.4, [14]). *An update step $G$ is said to $\gamma$-bounded if for all $w \in \mathcal{W}$,*

$$\|G(w) - w\| \le \sqrt{\gamma}. \tag{105}$$

**Lemma A.9** (Lemma 2.5, [14]). *Let $G_1, \ldots, G_T$ and $G'_1, \ldots, G'_T$ be two arbitrary sequences of updates. Let $w_0 = w'_0$ (same initial weights), and let $\delta_t = \|w_t - w'_t\|$ where $w_t$ and $w'_t$ are defined recursively through $w_{t+1} = G_t(w_t)$ and $w'_{t+1} = G'_t(w'_t)$. If $G_t$ and $G'_t$ are $\gamma$-bounded, then*

$$\delta_{t+1} \le \delta_t + 2\sqrt{\gamma}. \tag{106}$$

Let $G_1, \ldots, G_T$ be the sequence of updates made when SGD is used with the dataset $z_{-i}^{n-1}$, and let $G'_1, \ldots, G'_T$ be the sequence of updates made when SGD is used with the dataset $z_{-j}^{n-1}$. If the previous updates are $\gamma$-bounded, then Lemma A.9 gives us a bound on $\|w_{-i} - w_{-j}\|$ if we use $T$ iteration of SGD, and we obtain the desired result.

## B Experimental Details and Additional Experiments

We fine-tune an ImageNet pre-trained ResNet-18 model on MIT67 and Oxford Pets datasets. In order to compute the information bounds we use the results in corollary 4.1. We compute $w_{-i}/h_{-i}$ by removing sample $i$ from the training set and re-training from scratch. For all the experiments we remove 10 samples one at a time across 3 random seeds and use corollary 4.1 to compute the information bounds. We used 2 NVIDIA 1080Ti GPUs and the experiments take 1-2 days. We also include the results of fine-tuning on the Caltech256 dataset [10] (with ≈30000 samples).

# C Societal Impact and Limitations

## C.1 Societal Impact

Our work is of a theoretical nature, and so there is no direct social impact. However applications of deep neural networks, if not done properly, could have several consequence in society, including in terms of fairness, ability for misinformation etc. However, theoretical work that attempts to understand & predict the performance of deep neural networks could give tools to ameliorate these issues.

## C.2 Limitations and Future Work.

There are several open questions that remain from this work, pointing to some of its limitations. Natural ones include tighter bounds on the generalization gap, obtaining impossibility results in terms of error performance among several others.