# OpenReview forum: "On Leave-One-Out Conditional Mutual Information For Generalization"
_NeurIPS.cc/2022/Conference — NeurIPS 2022 Accept_

### Official Review · Reviewer_FLYU · 2022-07-01

**Rating:** 6
**Confidence:** 4
**Soundness:** 3 good
**Presentation:** 3 good
**Contribution:** 3 good

**Summary:**

Several generalization bounds have been derived in the literature for machine learning models using some notion of mutual information. These differ in how to measure the mutual information (e.g. whether it is between the hypothesis and the entire sample S or a subset of it) and whether or not the mutual information is conditional on other random variables and so on. In this work, the authors present yet another approach in which a single example is removed from the training set. The authors argue that the primary advantages of this approach are that it can be computed in O(n) time (since there are n examples to remove from) and that it is non-vacuous. They further show that measuring the mutual information between the predictions of the model and the data also yields similar bounds, which are tighter by the data processing inequality. Finally, the authors relate such bounds to stability and flatness.

**Questions:**

- Please double check the bound in Lemma 4.1

**Limitations:**

Please see the comments above for the limitations of the work. Since this is a theoretical work, there are no societal impact as far as I can see.

**Strengths And Weaknesses:**

*Strengths*:

1- The paper introduces a new approach for analyzing the generalization gap that can be useful to the community.

2-  The authors connect it nicely to related concepts, such as stability and flatness.

*Weaknesses*:

1- There are no tightness results. In fact, the bounds seem to be vacuous in most settings.

2- The SGD bound is odd because the bound increases with the number of SGD steps T, when a reasonable bound should be decreasing with T.

**Details**

The paper introduces a new information-theoretic approach for bounding the generalization gap of machine learning models. The paper is well-written and I think it offers great insight. For example, it suggests a new approach for quantifying flatness of the local minima that does not suffer from pitfalls observed in prior works (e.g. that one can arbitrarily change it via re-parameterization without changing the decision boundary). I think it would strengthen the paper to apply this measure of flatness on techniques that arguably improve flatness (e.g. by comparing it for neural networks trained with or without skip connections, SAM optimizer [1] or by applying it to methods trained via reinitialization [2, 5]).

On the downside, the authors emphasize that their approach yields non-vacuous bounds. However, the form of the bounds in Corollary 4.1 does not really support that argument. If the quadratic term in the exponent is bounded away from zero by a small epsilon, the entire bound is O(log n), which is vacuous for large training sets. For SGD, it seems strange to me that the bound increases with T, when it should decrease.

In the experiments, the authors use small datasets with a few thousand training examples. At minimum, the authors should show that larger training sets yield better bounds as they should given the interpretation in terms of stability. For example, they can train on subsets of the data and report results as the subset size increases. Also, the authors use pretrained models and fine-tune for a few epochs. These are (by nature) stable since pretraining improves stability. I think the authors should include experiments without pretraining to show that they still obtain useful (non-vacuous) bounds. The reason I am emphasizing on this is that this has been a main motivation throughout the paper so it should be demonstrated experimentally.

Overall, I still think the paper is worthwhile and I hope that the authors would address the issues above in the experiments to make their argument more convincing.

Some minor comments:

1- The technique of introducing noise to control the bound has been used in PAC-Bayes methods and it has been connected to flatness as well. See for example [3] and [6].  Perhaps, this should be mentioned in the related works. It would be useful to include some of those bounds in the comparison.

2- In Section 5, the authors claim that the earliest works were published in 2016. There are earlier works that derive non-vacuous generalization bounds with tightness results (see for instance [4]), in which the mutual information is measured between the hypothesis h and a single training example. In the latter case, the bounds are non-degenerate even for deterministic algorithms with uncountable hypothesis spaces. It has also been connected to stability, like in the present work.

3- Typo in Definition 4.2: "we say that $\beta_1$ test stability if".

[1] Foret, P., Kleiner, A., Mobahi, H. and Neyshabur, B. "Sharpness-aware minimization for efficiently improving generalization." ICLR, 2021.

[2] Alabdulmohsin, I., Maennel, H., & Keysers, D. (2021). The impact of reinitialization on generalization in convolutional neural networks. arXiv preprint arXiv:2109.00267.

[3] Neyshabur, B., Bhojanapalli, S., McAllester, D. and Srebro, N., "Exploring generalization in deep learning." NeurIPS, 2017.

[4] Alabdulmohsin, I. "Algorithmic stability and uniform generalization." NeurIPS, 2015.

[5] Zaidi, S., Berariu, T., Kim, H., Bornschein, J., Clopath, C., Teh, Y.W. and Pascanu, R., 2022. When Does Re-initialization Work?. arXiv preprint arXiv:2206.10011.

[6] Dziugaite, G.K. and Roy, D.M., 2017. Computing nonvacuous generalization bounds for deep (stochastic) neural networks with many more parameters than training data. arXiv preprint arXiv:1703.11008.

---

> ### Author Response · Authors · 2022-08-02
> **Response To Reviewer FLYU**
>
> We thank the reviewer for the care they took in reading our work and for their helpful comments. We address individual comments as follows:
> > There are no tightness results. In fact, the bounds seem to be vacuous in most settings.
>
> We have shown numerically that the bound gives non-vacuous results on the generalization gap in common deep learning settings. In addition, we show in Section 4.3 how we can obtain the standard bound generalization given uniform stability assumptions.
>
> > On the downside, the authors emphasize that their approach yields non-vacuous bounds. However, the form of the bounds in Corollary 4.1 does not really support that argument.
>
> We agree that the expression in (18) might give this initial impression. However, if we plug in $\epsilon$ for the quadratic term $(w_{-i} - w_{-j})^T \Sigma^{-1} (w_{-i} - w_{-j})$ in the exponent of the bound in Corollary 4.1, then we obtain:
> $$\ln(n) -\frac{1}{n}\sum_{i=1}^{n} \ln \left(ne^{-\epsilon}\right) = \ln(n)  - \ln(ne^{-\epsilon}) = \epsilon.$$
> The value of the bound (when moving the quadratic term by an epsilon) does not depend on the number of samples in the dataset. We will make sure to discuss this in the paper and possibly write the bounds of Corollary 4.1 differently in order to convey this.
>
> > The SGD bound is odd because the bound increases with the number of SGD steps T, when a reasonable bound should be decreasing with T.
>
> We thank the reviewer for pointing out a possible source of confusion in interpreting Lemma 4.1, which we will clarify in the paper. The SGD bound is obtained by using our loo-CMI bound together with the stability analysis in eq. 18 in order to bound the distance between the final weights when training with datasets that differ by one example. The training algorithms start from the same initialization and the distance between outputs (weights) grows larger with more iterations of SGD (as seen in Fig. 1). This result relates directly to the observation that early stopping (training for a smaller number of steps T) may improve the generalization of the network. Moreover, we point to [A], where authors show that models trained with SGD for a reasonable amount of steps, attain a small generalization gap.
>
> >In the experiments, the authors use small datasets with a few thousand training examples. At minimum, the authors should show that larger training sets yield better bounds as they should given the interpretation in terms of stability. For example, they can train on subsets of the data and report results as the subset size increases. Also, the authors use pretrained models and fine-tune for a few epochs. These are (by nature) stable since pretraining improves stability. I think the authors should include experiments without pretraining to show that they still obtain useful (non-vacuous) bounds. The reason I am emphasizing on this is that this has been a main motivation throughout the paper so it should be demonstrated experimentally.
>
> We compute the bounds when fine-tuning large networks pre-trained on ImageNet because (1) this is one of the most common real-world settings in deep learning, and (2) to validate the bounds and the analysis. Given the limited time of the rebuttal, we are able to provide results on Caltech256 (with ~30000 samples) at this time. We can add more experimental results in future versions of our paper.
>
> | Dataset | Train Error | Test Error | floo-CMI|
> | :---        |    :----:   |          ---: | ---: |
> | Caltech256[B] | 0      |0.2523      | 0.5807  |
>
>
>
> We also thank the reviewer for pointing out the references. In particular, [4] in the review does precede the earlier works cited in our “related works” section. We will make sure to have a comprehensive list of references and remove typos in our paper.
>
> [A] Moritz Hardt, Ben Recht, and Yoram Singer. Train faster, generalize better: Stability of stochastic gradient descent. In International conference on machine learning, pages 1225–1234. PMLR, 2016.
>
> [B] Griffin, Gregory & Holub, Alex & Perona, Pietro. (2007). Caltech-256 Object Category Dataset. CalTech Report.

---

> > ### Comment · Reviewer_FLYU · 2022-08-04
> > **Follow up**
> >
> > Thank you for the response. This clarifies my question about the tightness of the bound in (18), which was my main concern.
> >
> > Regarding Lemma 4.1, I suggest that you move the definition of $\gamma$ boundedness to the main paper. I see now that $\gamma$ relates to the learning rate.
> >
> > One additional comment, the appendix does not have the correct theorem numbers (e.g. Lemma 4.1 is written as Lemma A.7). Also, the statement of the lemma in the appendix is different from the main paper ($T^2$ vs. $T$).
> >
> > **Question**: Does the result on Caltech256 also use an ImageNet pretrained model or is it trained from scratch?

---

> > > ### Author Response · Authors · 2022-08-04
> > > **Response To Follow up**
> > >
> > > Thank you for your comment. We are glad to have been able to address your main concern.
> > >
> > > > Regarding Lemma 4.1, I suggest that you move the definition of boundedness to the main paper. I see now that relates to the learning rate.
> > >
> > > You are correct in saying it is related to the learning rate. It is also related to the boundedness of the gradient (with respect to a norm). We agree that it would be better for readability to include a precise definition of the boundness of the SGD update steps prior to the statement of Lemma 4.1.
> > >
> > > > One additional comment, the appendix does not have the correct theorem numbers (e.g. Lemma 4.1 is written as Lemma A.7). Also, the statement of the lemma in the appendix is different from the main paper (T vs. T^2).
> > >
> > > Indeed this is a typo. While it doesn’t change the implications of the bound, we apologize and have since corrected it in the supplementary (which also includes the full paper).
> > >
> > > > Does the result on Caltech256 also use an ImageNet pretrained model or is it trained from scratch?
> > >
> > > Due to the limited time for the rebuttal and discussions, we opted for a pretrained model. In using Caltech256, we wanted to be able to provide the reviewer with numerics on dataset of a larger scale than the datasets included in the main paper.

---

### Official Review · Reviewer_wWYy · 2022-07-10

**Rating:** 8
**Confidence:** 2
**Soundness:** 3 good
**Presentation:** 4 excellent
**Contribution:** 4 excellent

**Summary:**

The paper considers conditional mutual information based generalisation bounds for understanding the generalisation properties of neural networks. It builds on techniques introduced by prior work and overcomes a practical limitation: prior work requires computations that grow exponentially in dataset size, while the proposal entails only linear growth in dataset size.

**Questions:**

See above

**Limitations:**

See above

**Strengths And Weaknesses:**

The paper is very well written. I am not an expert in this area (hence also low confidence in my review) but the paper was a pleasure to read.

The paper is contributing to understanding generalisation in deep neural networks. This is one of the most important problems in the ML community today, and I welcome and support work towards this.

Since I am not an expert, I cannot fully evaluate the novelty of the work. However, if the authors are correct in their description of the current state of the field, I believe their results are very impactful.

Nonetheless, I have a few comments and things that i think could improve the paper:



- please use the actual submission template with line numbers, it makes it much easier to review
- typo page 2, paragraph 2, line 3: our loo-CMI only needS to
- theorem 4.1: I didn't understand what U' brings. In equation 16, doesn't the independence of U' mean that  p_{w|zn, u'} = p_{w|zn} ? Or do you mean "independent" in the sense of "another" or "duplicate" rather than the statistical sense?
- could you comment on the computational requirements of this method? You need to train N models, one for each datum right? This must be very slow for reasonably sized datasets.
- One idea for evaluating your method on larger datasets: practitioners sometimes fine-tune just the last layer of a DNN, rather than the whole network. If you precompute features of the data up to this last layer, this is like training a linear classifier. This could enable you to try out your method on larger datasets.

---

> ### Author Response · Authors · 2022-08-02
> **Response To Reviewer wWYy**
>
> We thank the reviewer for reading our paper and making insightful suggestions. We address specific comments and questions as follows:
>
> * We have added line numbers.
> * We thank the reviewer for pointing out our typo. We will make sure to remove all typos for the final version.
> * U’ is a random variable with the same distribution as $U$, but independent from $U$. In other words, $U$ and $U’$ are iid. In theorem 4.1, $p_{w| z^n, u'} = p_{A(z_{-u'}^{n-1}) |z^n, u'}$. Indeed, this is confusing as one cannot determine that $W$ depends on $U’$ from the notation used. We have changed the notation of Theorem 4.1 to avoid this by denoting $\mathcal{A}{z^{n-1}_{-u'}}$ by $W'$.
> * We use NVIDIA 1080Ti GPU to run the experiments and the experiments take 2 days.
> * Indeed, you are correct. We compute the bounds by training neural networks that are pre-trained on a different dataset (in our case, ImageNet). We compute the bounds when fine-tuning pre-trained networks since it is one of the most common settings in real-world applications of deep learning. Moreover, this also enables faster convergence and hence faster computation of the bound. We can take it a step further as you have suggested, and only train the last layer of the network. In particular, we note that in this case if we use an L2 loss to train, we can compute all the LOO results at the same time in one step https://dspace.mit.edu/handle/1721.1/37318. We will discuss this in future versions of the paper.

---

### Official Review · Reviewer_dogW · 2022-07-11

**Rating:** 7
**Confidence:** 2
**Soundness:** 4 excellent
**Presentation:** 4 excellent
**Contribution:** 3 good

**Summary:**

This paper argues that standard conditional mutual information (CMI) bounds are difficult to compute and interpret. Inspired by the spirit of leave-one-out cross-validation, the authors present a computable and interpretable bound termed loo-CMI. In particular, the proposed loo-CMI assumes that a generalized algorithm should not be affected by a small perturbation (i.e., leaving one training sample out at random). Furthermore, they interpret their bound in the function space, which leads to a tighter bound (i.e., floo-CMI). Empirically, they evaluate the quality of the bound and provide sufficient analysis.

**Questions:**

1. Does the small number of samples result in high randomness of the previous empirical results? Did the authors do ablation studies on ``the number of samples'', especially for the larger number?
2. How does the pre-training dataset size influence the quality of the proposed bounds?

**Limitations:**

Yes, the authors have addressed the societal impact of their work.

**Strengths And Weaknesses:**

This idea is intrigued by an intuitive assumption that a generalized algorithm should not be affected by a small perturbation. The authors do a good job of motivating this work, and the proposed bounds (i.e., loo-CMI and floo-CMI) are simple but reasonable. Also, the paper is very well written and easy to understand.

My main doubts/concerns regarding the paper are the following:

- In Fig 2 and 3, there is a big difference between the main paper and supplementary material. Does it mean that the previous empirical results suffer from high randomness when the number of samples is small? Did the authors do ablation studies on ``the number of samples'', especially for the larger number?
- While the analysis of the dataset size during fine-tuning is interesting, I believe the pre-training dataset plays a more crucial role in determining the generalization ability of deep models. Can the authors compare their results on different pre-training dataset scales?

Minor:

- The code should mention the key lines corresponding to loo-CMI and floo-CMI.
- The Conclusion section needs to be added.

---

> ### Author Response · Authors · 2022-08-02
> **Response To Reviewer dogW**
>
> We thank the reviewer for reading our paper and making helpful suggestions. We address the comments as follows:
>
> > In Fig 2 and 3, there is a big difference between the main paper and supplementary material. Does it mean that the previous empirical results suffer from high randomness when the number of samples is small? Did the authors do ablation studies on ``the number of samples'', especially for the larger number?
>
> We think the randomness is partly due to the varying informativeness of different samples i.e. some samples contribute more in determining the final weights of the network compared to others. There is prior work on measuring the informativeness of a single sample in a dataset [A] (see below) (reference [11] in our paper). Here, informativeness is measured relative to the rest of the training dataset. Informative samples are expected to have a larger effect on the output of the algorithms compared to less informative samples (e.g. samples that are very similar to other samples in the dataset).
>
> > While the analysis of the dataset size during fine-tuning is interesting, I believe the pre-training dataset plays a more crucial role in determining the generalization ability of deep models. Can the authors compare their results on different pre-training dataset scales?
>
> We compute the bounds when fine-tuning large networks pre-trained on ImageNet as these networks are freely available on standard libraries and one of the most common settings in real-world applications of deep learning. Such networks require days and sometimes weeks to train from scratch. However, we agree with this point and think it is worthwhile to compute the bounds in the case of different pre-training datasets for future versions of the paper.
>
> > The code should mention the key lines corresponding to loo-CMI and floo-CMI.
>
> We will update our code so that the lines of code pertaining to loo-CMI and floo-CMI are easy to find.
>
> >The Conclusion section needs to be added.
>
> We will make sure to add a conclusion to our paper.
>
> [A] Hrayr Harutyunyan, Alessandro Achille, Giovanni Paolini, Orchid Majumder, Avinash Ravichandran, Rahul Bhotika, and Stefano Soatto. Estimating informativeness of samples with smooth unique information. In International Conference on Learning Representations, 2021.

---

### Official Review · Reviewer_tDeb · 2022-07-11

**Rating:** 5
**Confidence:** 3
**Soundness:** 3 good
**Presentation:** 3 good
**Contribution:** 2 fair

**Summary:**

This paper presents a new leave-one-out conditional mutual information method to bound generalization risk. Specifically, two information theoretic generalization upper bounds, namely loo-CMI and floo-CMI, are proposed. The authors claim compared to standard CMI, the proposed bounds are more cost-effective because they reduce the computing cost from 2^N to only N. Strong theoretical analysis are provided. In addition, some empirical results have been reported in image-classification problem.

**Questions:**

1. Can these bounds be used for model training? Or they only can be used to do model selection.
How costly does it take to compute the bounds?

2. It is puzzling that the results in Table 1 all have training error to be 0. Do they imply the models have overfitted the data?


**Ethics Review Area:**

["I don’t know"]

**Limitations:**

My major concern is the utility of the methods, since the authors do not show any actual applications of their bounds to facilitate the real-world machine learning practices.

**Strengths And Weaknesses:**

Strengths:
1.	The authors proposed the novel information generalization bounds.
2.	The theoretical analysis is sound, and it shows the bounds have good properties.


Weakness:
1.	The authors should compare their methods to competing baselines. For example, the f-CMI bound from paper [1]. Moreover, the author should provide the results for more complex datasets, such as ImageNet.
2.	In Figure3, the authors should plot training errors+ proposed generalization bounds, do they dominate the test error? In addition, the stability analysis is missing. And the authors should repeat the experiments and report the error bar to show the variance of the bounds in Figure 2&3.
3.	The authors need to demonstrate their methods actually works in empirical settings. More concretely, the author should apply their methods on a set of different models trained on the same real-word dataset and show the predicted generalization bounds are consistent with the actual test error.

[1] Harutyunyan, Hrayr, et al. "Information-theoretic generalization bounds for black-box learning algorithms." Advances in Neural Information Processing Systems 34 (2021): 24670-24682.

---

> ### Author Response · Authors · 2022-08-02
> **Response To Reviewer tDeb**
>
> We thank the reviewer for their detailed and careful reading of the paper and the constructive comments. We address specific comments as follows:
>
> > The authors should compare their methods to competing baselines. For example, the f-CMI bound from paper [1]. Moreover, the author should provide the results for more complex datasets, such as ImageNet.
>
> Unfortunately, due to the random subset setting considered in [1] (exponential in the number of samples), the computational time required does not allow obtaining a meaningful comparison in the limited rebuttal and discussion period. We do however note that the final form of the bounds is similar with the main difference that our bound depends on changing a single sample and only has a linear number of terms. This generally allows us to exploit the stronger stability of SGD when changing only one sample and provides a tractable computation of the bound.
>
> However, we add results on a larger dataset: Caltech256 [A] (see below) with ~30,000 samples.
> | Dataset | Train Error | Test Error | floo-CMI|
> | :---        |    :----:   |          ---: | ---:|
> | Caltech256[A] | 0      |0.2523      | 0.5807   |
>
> > In Figure3, the authors should plot training errors+ proposed generalization bounds, do they dominate the test error? In addition, the stability analysis is missing.
>
> We refer the reviewer to Table 1 and Fig. 2, where we compare the actual test error and our generalization bound (in Fig. 2, generalization gap ~= test error since we train to zero training error). The aim of Fig. 3 is instead to show the tradeoff between the two quantities in Eq. 19 when $\sigma$ changes. If we understand the reviewer’s second comment correctly, we refer to Section 4.5 for an analysis on stability and the proposed bounds.
>
> > And the authors should repeat the experiments and report the error bar to show the variance of the bounds in Figure 2&3
>
> We have added error bars in the figures of the supplementary. We refer the reviewer to (Figures 4,5 in Section B of the supplementary).
>
> > The authors need to demonstrate their methods actually works in empirical settings. More concretely, the author should apply their methods on a set of different models trained on the same real-word dataset and show the predicted generalization bounds are consistent with the actual test error.
>
> We are showing empirical results in real-world settings and datasets with large images(ResNet-50 trained on MIT-67, Ox. Pets) in Table 1 and Fig. 2. Unfortunately, due to the limited time in the rebuttal, we are unable to provide additional experiments currently. We can include more numerical results in future versions of the paper.
>
> > Can these bounds be used for model training? Or they only can be used to do model selection. How costly does it take to compute the bounds?
>
> Besides model selection, observations from the bounds derived in the paper can be used to suggest new training algorithms to improve generalization. In particular, one implication to be explored for future work, is a form “weighted” gradient clipping scheme based on the expressions of Corollary 4.1(where the goal of the algorithm is to implicitly minimize the bounds). Another direction is to modify the training algorithm to implicitly minimize the re-normalized flatness defined in Sect. 4.5. These form future implications of our current work.
>
> >It is puzzling that the results in Table 1 all have training error to be 0. Do they imply the models have overfitted the data?
>
> We have used large over-parameterized neural networks in our simulations such as ResNet-18. Deep models have been shown to fit the data perfectly (classically overfit) while still exhibiting low test error and good generalization (see below) [B].
>
> [A] Griffin, Gregory & Holub, Alex & Perona, Pietro. (2007). Caltech-256 Object Category Dataset. CalTech Report.
> [B] Belkin, M., Hsu, D., Ma, S., and Mandal, S. Reconciling modern machine-learning practice and the classical bias–variance trade-off. PNAS, 116:15850–15854, 2019

---

### Meta-Review · Area_Chair_AF9o · 2022-08-22

**Recommendation:** Accept
**Confidence:** Certain

**Metareview:**

The paper derives an information theoretic generalization bound based on a mutual information measure that relies on a leave-one-out procedure to reduce the cost of previous estimators. This estimator reduces the complexity from exponential ($2^N$ where $N$ is the sample size) to linear.

Overall, the general feeling among the reviewers and myself is that the paper does make a rather novel contribution, which is backed up by theoretical bounds. I think the reduction in terms of complexity from  to  is significant, but one could still argue this is not quite sufficient yet for modern tasks in machine learning, where  can be very large, so  evaluations of a large neural network can still be very costly. This aspect seems a bit hidden in the paper where the experiments are only on very small datasets. The authors do not even seem to explicitely mention the size of the datasets.

Still, I think the theoretical contribution of the paper is significant enough so I recommend acceptance but I stronlgly encourage the authors to further discuss the computational aspect.

I would like to point out two additional relevant work:

1) https://arxiv.org/pdf/2206.14800.pdf
This paper seems to develop very similar bound also using a concept of mutual information and LOO.

2) https://arxiv.org/pdf/2203.03443.pdf
This paper reduces the computational aspect of LOO, although it seems to be only applicable to kernels.

**Award:**

No

---

### Decision · Program_Chairs · 2022-09-14

Accept